# Evaluation Method for Urban Public Service Carrying Capacity (UPSCC): A Qualitative–Quantitative Bi-Dimensional Perspective

**DOI:** 10.3390/ijerph182312539

**Published:** 2021-11-28

**Authors:** Shiju Liao, Xiaoyun Du, Liyin Shen, Minghe Lv

**Affiliations:** International Research Centre for Sustainable Built Environment, School of Management Science and Real Estate, Chongqing University, Chongqing 400000, China; liaoshiju@cqu.edu.cn (S.L.); Shenliyincqu@163.com (L.S.); 202103041051t@cqu.edu.cn (M.L.)

**Keywords:** urban public service carrying capacity (UPSCC), qualitative UPSCC, quantitative UPSCC

## Abstract

Urban Public Service Carrying Capacity plays an essential role in urban social and economic development. However, existing study has been focused on the evaluation of UPSCC from a quantitative perspective. It is necessary to evaluate UPSCC from a qualitative–quantitative bi-dimensional perspective. This paper establishes an innovative evaluation method for UPSCC based on a qualitative–quantitative bi-dimensional (QQBD) perspective. The proposed QQBD-based UPSCC evaluation method can help identify the weak areas of public services. The conclusions of this study are as follows. Firstly, public services are people-oriented social resources, which should be evaluated from both quantitative and qualitative perspectives. Secondly, the quantitative measurement of public service carrying capacity needs to consider both UPSCC load and carrier, while the qualitative measurement needs to consider the satisfaction among stakeholders. Thirdly, the demonstration of the case study cities shows the effectiveness of the qualitative–quantitative bi-dimensional UPSCC evaluation method. By applying the QQBD-based UPSCC evaluation method introduced in this study, decision makers can identify the specific areas that affect the UPSCC performance, and thus tailor-made policy can be designed for improving UPSCC performance by adjusting UPSCC quantity and quality.

## 1. Introduction

Urban public services are essential resources for supporting socioeconomic activities which provide essential education, medical treatment, culture and sports facilities, social welfare and security, and other basic services to urban residents [1,2]. Many studies have found that urban public services exhibit a notable externality that can improve the satisfaction and happiness of residents [3,4]. Quality public services provided by cities, especially in mega cities, play a significant role in constantly attracting floating populations in the course of rapid urbanization [5,6]. It is estimated that 3.176 billion people have become urban residents worldwide during the period of 1960–2018 as countries have been moving from agricultural to industrialized economies [7]. Urban challenges induced by population inflows in the fast urbanization process exert tremendous pressure on the capacity of public services, and these typical challenges include urban environmental degradation [8], traffic congestion [9,10], housing shortage and unaffordable housing prices [11,12], urban unemployment [13], inequality of public health [14], and inequity between rich and poor [15]. These challenges provoke a reflection of whether urban public services have the appropriate capacity to carry urban resilience and sustainability.

The carrying capacity of urban public services is a significant factor affecting urban sustainable development. On one hand, the capacity will support urban sustainable development if public service resources are developed and allocated properly to address the ever-growing demands from the public and to support socioeconomic growth [15,16]. In other words, proper carrying capacity of urban public services is essential to urban sustainable development. On the other hand, the carrying capacity of public service resources will restrain urban sustainable development if the capacity does not match with the urban development pattern, followed by various urban diseases and the deterioration of quality of public living environment.

This mismatch can be categorized into two types, namely, shortage of urban public service carrying capacity and irrational oversupply of public service resources. On one hand, if there is a shortage of such capacity, the urban social-economic function will be restrained. In fact, this appears to be an increasing phenomenon in many cities across the world. For illustration, Yin et al. (2018) points out that the carrying capacity of public medical resources in China cannot meet the rapidly increasing demand in the context of rapid urbanization, and it is common in China for patients to queue all night to register to see a doctor [14]. Zimbelman et al. (2010) evaluated the physical therapist shortage from 2008 to 2030 within all 50 states in the U.S., and the results suggested the number of states confronting acute physical therapist shortages would increase from 12 to 48 [17]. This indicates that the carrying capacity of the rehabilitation medical service will face a severe challenge in the United States. In conducting the study about the supply and demand of compulsory education in Chinese mega cities, Yang (2016) pointed out that the capacity of compulsory education resources in big cities is still unable to meet the expansion of school-age children flowing from small cities and villages, although the compulsory education resources have been increasing in these cities. On the other hand, resource waste will result if there is an irrational oversupply of public service resources. This actually happens in many cities as well [18]. Yang et al. (2021) demonstrated that the average sickbed use rate of primary medical institutions was only 75% between 2011 and 2015 in China [19]. In a study in Perth, Australia, Billie and Robert (2002) found that residents had good access to recreational facilities, but used them less, suggesting either that preferences were at odds with offerings, or that other types of access barriers intervened to lower use rates, resulting in a waste of public recreation resources [20]. In fact, neither shortage nor oversupply of public service resources is conducive to urban sustainable development. It is, therefore, essential to properly evaluate the state of urban public service carrying capacity (UPSCC) to find out whether the public service supply matches with the demand for a specific city. The understanding of the state of UPSCC can help make proper decisions to coordinate the UPSCC with socioeconomic activities towards urban sustainable development.

Several studies have conducted an evaluation of the carrying capacity of urban public service in particular cities. Yin and Liu (2017) evaluated the UPSCC of Beijing–Tianjin–Hebei region in China by using the data surveyed for 2016, and the results suggested the carrying capacity of public service in Hebei was significantly lower than the other two regions [21]. Wei and Wang (2019) examined the UPSCC of Shanghai in China for the period of 2005–2016 by applying the P-S-R model, and clarified the state of UPSCC of different districts into three types, namely, heavy overloading, overloading, and full-loading [22]. Wang et al. (2018) analyzed the growing trend and regional disparity of UPSCC in four mega cities in China by using the data surveyed for the period of 2005–2015, and found that the four cities were obviously different in the level, structure and changing tendency of UPSCC [23].

Previous studies relating to evaluation of UPSCC have mostly focused on the quantitative discussion of whether the carriers (public services) have sufficient carrying capacity (generally the scale and quantity) to carry the load of human activities. However, different from natural resources and the environment, public services are inextricably linked to urban residents’ daily lives. The fact that quality perceptions have a strong influence on citizens’ inclination to avail themselves of public services is beyond dispute. In the case of the same scale of public services supply, the UPSCC will have significant variation due to the different internal supply structure and the pattern of urban development. It is insufficient to assess UPSCC only in terms of the number and scale of public services provided by the government, in other words, quantitative assessment. For example, a research report released by Center for Development Research of Bangladesh indicated the proportion of GDP allocated to the public health care sector between 1985/86 and 1995/96 had more than doubled compared to the past decade, and 346 private hospitals were established in the period of 1982–1996 [24]. However, the improvements in allocation and access did not meet the actual demand because of urban residents’ negative perceptions of service quality, resulting in a continuation of low use of the facilities and an arresting phenomenon of seeking health care services in neighboring countries for those who can afford it [25]. Yu et al. (2015) surveyed the data of the demand for and use of medical services from 1993 to 2008 in China and found that the medical services supply was growing; meanwhile, the demand showed an unexpected decreasing trend. Judging by the quantity index alone, it seemed that the carrying capacity of public medical resources had improved [26]. However, there exists extensive potential medical demand that is suppressed by the rapid growth of medical expenses and the unreasonable structure of China’s health delivery system [26]. Therefore, evaluation of UPSCC from a quantitative perspective only ignores the problems behind the numbers. It is necessary and of great significance to evaluate UPSCC from a qualitative–quantitative bi-dimensional perspective. Without this qualitative–quantitative perspective, the UPSCC assessment would be inappropriate, and the policy instruments designed may accordingly be improper.

For the reasons discussed previously, this paper aims to fill the gap in the field of UPSCC by establishing an innovative evaluation method for UPSCC based on qualitative–quantitative bi-dimensional perspective. In line with the research aim, the overall structure of the study takes the form of five specific research modules: (1) to design the methodology of this study (Section 2); (2) to develop the qualitative–quantitative bi-dimensional (QQBD) evaluation indicators for UPSCC (Section 3); (3) to develop the qualitative–quantitative bi-dimensional (QQBD) evaluation model for UPSCC (Section 4); (4) to demonstrate the effectiveness of the established QQBD model by means of a Chinese context-based case study (Section 5 and Section 6); and (5) to draw the conclusion of this study (Section 7).

## 2. Methodology

To achieve the research aim, the overall research process of this study was designed as shown in Figure 1. The research framework includes four procedures, namely, literature survey, development of qualitative–quantitative bi-dimensional (QQBD) evaluation model for UPSCC, development of analysis framework for evaluation results, and demonstration of QQBD model application.

Firstly, a comprehensive literature review was conducted to clarify the definition and connotation of UPSCC and to identify the dimensions included in a public service system, which can guide the subsequent construction of the index system of UPSCC.

Secondly, a qualitative–quantitative bi-dimensional model was established for evaluating the state of UPSCC. This procedure includes two components: construction of an indicator system and establishment of an evaluation model for UPSCC. Based on a review of the indicators that previous studies have used to investigate the performance of public services, this paper formulates a quality index and quantity index for measuring the qualitative and quantitative carrying capacity of urban public services, respectively. Then, appropriate evaluation models are developed based on the discussion of the advantages and disadvantages of previous qualitative methods and quantitative approaches for evaluating the performance of UPSCC.

In the third research procedure, a qualitative–quantitative bi-dimensional analysis framework is established to analyze the evaluation results of UPSCC based on Boston Matrix technique, and specific cities are further distinguished into four categories according to their qualitative and quantitative level of UPSCC.

In the final procedure, a demonstration is provided to show the application of QQBD model for evaluation of UPSCC. Thirty-five mega cities in China were selected as the case cities, and the effectiveness of the QQBD model is investigated and analyzed. Based on the case study, suggestions and policy recommendations are proposed to improve the carrying capacity of urban public services, and to apply the evaluation model effectively in practice.

## 3. Development of Qualitative–Quantitative Bi-Dimensional (QQBD) Indicators for Measuring UPSCC

### 3.1. Definition and Connotation of UPSCC

Carrying capacity (CC) was initially an engineering geological term, referring to the maximum load that a carrier can carry without causing any physical damage [27]. Since the 1790s, the concept of CC has been used by scholars in ecology and biology disciplines to study the carrying capacity of livestock in a specific pasture [28]. In 1798, Malthus’s population theory initially extended the concept of CC to human society [29]. In the 1960s and 1970s, with the outbreak of global problems such as resource depletion and environmental degradation, the research into CC was significantly extended to different particular fields, which can be summed up as single-factor carrying capacity (such as land, atmospheric environment, water resources, water environment, tourism, forest), and multiple-factor comprehensive carrying capacity (such as ecological carrying capacity, urban carrying capacity, environment carrying capacity, resources environment carrying capacity).

The literature contains various studies on the subject of urban carrying capacity (UCC), but there is little discussion about urban public service carrying capacity (UPSCC). Onishi and Takashi (1994) examined the capacity of the inner-city area of Tokyo based on the assumption that the citizens should lead comfortable lives making use of various urban facilities and services [30]. The results suggested that the living and working population in Tokyo already exceeds the limit for comfort, especially in terms of the congestion of commuter trains and roads, the ability for waste disposal, the prevention of air pollution, and the supply of affordable housing. Although the concept of UPSCC was not formally introduced in this study, it was embedded in it. Since then, UPSCC has not been given enough attention, and public services only appear as one of the dimensions of indicator systems in some studies regarding the evaluation of the urban comprehensive carrying capacity [31,32]. In 2016, Wang (2016) initially clarified the theoretical definition and component of UPSCC. The UPSCC is defined as the maximum load or scale of population and socio-economic activities that can be carried by various public services within a certain time period and administrative boundary [33].

Previous studies have paid little attention to UPSCC. However, the existing discussions of CC in the context of urban management could provide some inspiration when exploring the definition and connotation of UPSCC. The representative definitions of CC in the context of urban management are listed in Table 1.

It can be seen from Table 1 that the concept of carrying capacity in the context of urban management basically inherited the two core connotations of CC in engineering geology, namely the “maximum load or limit of load the carrier can carry” and “without causing serious degradation and irreversible damage”. However, as the city is an open, adaptive, nonlinear, resilient, and complex system [39,40], exploration of the limit of urban carrying capacity needs to be placed under a certain time, geographical scope and certain socio-economic development level. Furthermore, this limit or maximum is a potential existence, and it is hard to figure out a specific value in an open and resilient urban system [41].

Based on the above discussion and the characteristics of public services mentioned in the introduction section, UPSCC can be defined as: the maximum capacity of population and social-economic activities that can be supported by various public services without causing serious degradation and irreversible damage to urban system and citizen’s tolerance, over a certain period and socio-economic and technological development level. The definition of UPSCC contains four connotations: (1) there are two elements, carrier and load, contained in the concept of UPSCC. Carriers are the public service resources provided by the government, whereas loads represent various citizens’ demands and urban social-economic activities. UPSCC, in essence, is the interaction between the carrier (public service resources) and the load (population and social-economic activities); (2) there is a maximum or limit of UPSCC under particular time and social-economic conditions. However, this limit or maximum is a potential existence, as the city is an open and resilient system. (3) As exploring the maximum value of UPSCC, citizen’s tolerance is a significant parameter as it is difficult to define an exact destruction of urban system. (4) By using the terminology of elastic and plastic deformation from structural engineering, urban system resilience suggests that it is hard to state that there is a plastic or total deformation of an urban system unless under extreme climate change or war. For example, rising sea level resulting from climate change has presented a threat to the survival of Jakarta, the capital city of Indonesia, and Indonesia has decided to move the capital from Jakarta to East Kalimantan province [42,43]. Therefore, in the context of urban management, the exploration of the limit of UPSCC is more of a discussion based on elastic deformation.

### 3.2. Dimensions of UPSCC

Existing studies on the USPCC were collected by the literature review method, and typical USPCC dimensions were obtained, as shown in Table 2.

According to the existing dimensions of UPSCC in Table 2, seven high frequency dimensions are selected as the dimensions of UPSCC in this study: Basic Education (D1), Public Health (D2), Social Security (D3), Housing security (D4), Urban Environment (D5), Public Transportation (D6) and Public Culture and Sports (D7).

### 3.3. Indicators for Measuring Qualitative Carrying Capacity of Urban Public Services

It is very important to find out the qualitative indicators for measuring public service performance, and the data of qualitative indicators can be obtained through questionnaires with residents. This study adopts The Evaluation Report of Public Satisfactoriness Index for Basic Public Services in 38 Major Chinese Cities, 2019, which was published by the Chinese Academy of Social Sciences to reflect the quality of public service carrying capacity. A total of 14,345 questionnaires from Chinese cities were collected through an online questionnaire survey. The report includes the seven evaluation dimensions in this study, and the specific indicators for measuring the qualitative UPSCC performance. The specific process of the questionnaire can be seen in the report of The Evaluation Report of Public Satisfactoriness Index for Basic Public Services [62].

### 3.4. Indicators for Measuring Quantitative Carrying Capacity of Urban Public Services

#### 3.4.1. Carrier-Load Principle for Selecting UPSCC Quantitative Indicators

A carrier-load-based urban carrying capacity evaluation principle is adopted to select quantitative indicator for measuring quantitative performance of UPSCC. Previous studies have introduced some evaluation methods based on the carrier-load perspective for investigating urban resource environment carrying capacity by focusing on different types of urban resource. For example, Shen et al. (2020) proposed a new method for evaluating urban resource environment carrying capacity from the load-and-carrier perspective [41]. Liao et al. (2020) introduced a carrier-load perspective method for investigating the regional water resource carrying capacity [63]. Luo et al. (2020) developed a method for assessing urban land carrying capacity based on the carrier-load perspective [64].

#### 3.4.2. Selection of UPSCC Carrier-Load Indicators

The existing literature has proposed numerous indicators for evaluating the performance of UPSCC. A review of this literature can provide valuable references for establishing the indicators for measuring UPSCC carrier and load. Table 3 presents a variety of indicators for measuring the performance of UPSCC.

As can be seen from Table 3, existing studies have provided various indicators that can be used to measure the performance of UPSCC, but there is no distinction between load indicators and carrier indicators. In accordance with the discussion in the previous section, it is necessary to identify carrier and load indicators for measuring USPCC performance based on Table 3. In line with the carrier-load-based evaluation principle, carriers are the public service resources provided by the government, whereas loads represent various citizens’ demands and urban social-economic activities. By discussing the load and carrier properties of the indicator with experts, the carrier and load indicators of bearing capacity in seven dimensions were obtained in this study, as shown in Table 4.

#### 3.4.3. Calculation of UPSCC Quantitative Indicators Based on Carrier-Load Principle

As discussed in Section 3.4.1, both carriers and loads of UPSCC need to be considered collectively to properly investigate the carrying capacity of urban public services. The carriers of UPSCC should support the pressures from the loads. With respect to the carrying capacity formula in physics, the UPSCC quantitative indicators are considered to be a relative method, which is written as follows [41].
(1)$$X_{i}=L_{i}/C_{i}$$
where *X**_i_* refers to the value of UPSCC quantitative indicator *i*, *L_i_* refers to the value of UPSCC load indicators *i*, and *C_i_* refers to the value of UPSCC carrier *i*. Both the UPSCC load and carrier indicators are described in Table 4. According to this equation, UPSCC quantitative indicators can be obtained to reflect the interacted effect between carriers and loads.

## 4. Development of Qualitative–Quantitative Bi-Dimensional (QQBD) Evaluation Model for UPSCC

In this study, the entropy weight method was used to calculate the quantity score of UPSCC, and the equal weight method was used to calculate the quality score of UPSCC. Then, Boston matrix was used to construct a quality–quantity two-dimensional analysis model.

### 4.1. Establishment of Weighting Values between Indicators

#### 4.1.1. Weighting Method for Quantitative Indicators

Weight value is a very important parameter in comprehensive evaluation. Reasonable weight setting is the basis for accurately measuring the performance of UPSCC. Since the weight setting process is not affected by subjective factors, the weight distribution generated by the entropy weight method has higher effectiveness and objectivity, and the evaluation results are more accurate [65,66]. Therefore, the entropy weight method will be adopted in this study to set the weight of indicators.

According to existing research, there are three steps in the entropy weight method. Firstly, the value of indicator needs to be standardized; then, the entropy value of the index is calculated. Finally, the weight value of each indicator will be determined according to the entropy value. The specific calculation process is as follows [65].

a.Standardization of indicators

Firstly, Formulae (2) and (3) are used to standardize indicators, wherein Formula (2) applies to positive indicators that perform better with larger values, and Formula (3) applies to negative indicators that perform better with smaller values. All the indicators in this study are positive indicators, therefore Formula (2) is adopted.
(2)$$\gamma_{ij}=\frac{x_{ij}-Min_{j}\left\{x_{ij}\right\}}{Max_{j}\left\{x_{ij}\right\}\;-\;Min_{j}\left\{x_{ij}\right\}}$$
(3)$$\gamma_{ij}=\frac{Max_{j}\left\{x_{ij}\right\}\;-\;x_{ij}}{Max_{j}\left\{x_{ij}\right\}\;-\;Min_{j}\left\{x_{ij}\right\}}$$

In the above formula, the variable $x_{ij}$ is the original value of index *i* in the *j* th year and $\gamma_{ij}$ is the value of the standardized variable $x_{ij}$. In addition, $Max\left\{x_{ij}\right\}$ and $Min\left\{x_{ij}\right\}$ are the maximum value and minimum value of the original indicator values, respectively.

b.Calculate the entropy value of each indicator

Then, the weight value of each index is determined according to the standardized index, and the specific calculation formula is as follows.
(4)$$f_{ij}=\frac{\gamma_{ij}}{\sum_{j=1}^{m}\gamma_{ij}}$$
(5)$$k=\frac{1}{\ln m}$$
(6)$$H_{i}=-k\sum_{j=1}^{m}f_{ij}\ln f_{ij}$$
(7)$$w_{i}=\frac{\left(1-H_{i}\right)}{\left(n-\sum_{i=1}^{n}H_{i}\right)}$$
where *f_ij_* represents the proportion of indicator value *i* in a city to the sum of indicator value of all the cities, *m* is the number of evaluated cities, *n* is the number of evaluated indicators, *H_i_* represents the entropy value of indicator *i*, and *w_i_* represents the weight of indicator *i*.

#### 4.1.2. Weighting Method for Qualitative Indicators

As the data of quantitative UPSCC indicators are graded in a range of 0–100 with the same order of magnitude, the equal weight method is used in this study to calculate the weight of qualitative indicators, and the weight of all indicators is 1.

### 4.2. Linear Weighted Sum Method for Calculating Values of UPSCC Performance

The scores for the UPSCC performance in city *i* for year *j* (*y_ij_*) can be obtained by using linear weighted sum method according to following Formula (8).
(8)$$y_{ij}=\gamma_{ij}w_{i}$$

### 4.3. Boston Matrix Method for Analyzing QQBD-UPSCC Performance

This study used Boston matrix to classify the performance of qualitative UPSCC and quantitative UPSCC. According to the mean score of UPSCC quantity and UPSCC quality, the performance of UPSCC in each city can be divided into the following four quadrants, namely, quadrant I (high quantity, high quality), quadrant II (high quantity, low quality), quadrant III (low quantity, low quality), quadrant IV (high quantity, high quality), as shown in Figure 2.

## 5. Case Demonstration

### 5.1. Sample Cities

To demonstrate the effectiveness of the model proposed in the previous section, a case study including 35 large cities in China was conducted, and the spatial distribution of these sample cites is shown in Figure 3. In terms of geographical distribution, these 35 sample cities are located across the regions of northeast, east, central and western China, in order to better represent the regional variety in the country. From the perspective of socioeconomic performance, these 35 sample cities contributed to more than 36% of the national GDP and about 19% of the national population for the period from 2014 to 2018 [65]. Therefore, the selection of 35 sample cities is considered rational, important and representative in demonstrating the UPSCC performance in China. The general information of the case study cities is shown in Appendix A.

### 5.2. Calculation Results

#### 5.2.1. Qualitative Carrying Capacity of Urban Public Services

By referring to the report of The Evaluation Report of Public Satisfactoriness Index for Basic Public Services in 38 Major Chinese Cities, 2019, the qualitative UPSCC performance value of the 35 sample cities can be obtained. The qualitative UPSCC performance value for each of the dimensions D1, D2, D3, D4, D5, D6, and D7 and the overall performance value D are shown in Table 5.

As shown in Table 5, the mean scores of UPSCC qualitative performance in D1, D2, D3, D4, D5, D6 and D7 are 60.71, 65.25, 59.57, 50.37, 66.80, 62.11 and 64.92, respectively. It can be seen that the dimension with the best UPSCC qualitative performance is D5, and the dimension with the worst performance is D4. Furthermore, the value of overall UPSCC qualitative performance of 35 cities is 429.72. Eighteen of the 35 sample cities had better qualitative UPSCC performance than the average level in 2019: Xining, Lanzhou, Ningbo, Dalian, Xiamen, Guiyang, Shanghai, Hangzhou, Nanchang, Fuzhou, Yinchuan, Jinan, Qingdao, Taiyuan, Kunming, Hohhot, Changchun and Urumqi. The qualitative scores of UPSCC of the remaining 17 sample cities are lower than the average level, and the five bottom performers are Guangzhou, Shijiazhuang, Wuhan, Zhengzhou and Xi’an.

#### 5.2.2. Quantitative Carrying Capacity of Urban Public Services

By applying the data collected from 35 mega cities to Equations (2)–(8), the quantitative UPSCC performance value of the sample cities can be obtained. The quantitative UPSCC performance value for each dimension D1, D2, D3, D4, D5, D6, D7 and the overall performance value D are shown in Table 6.

As can be seen from Table 6, the mean scores of UPSCC quantitative performance in dimensions D1, D2, D3, D4, D5, D6 and D7 are 0.11, 0.12, 0.15, 0.04, 0.06, 0.09, 0.07 and 0.64, respectively. It can be seen that the dimension with the best performance of quantitative UPSCC is D3, while the worst dimension is D4. The mean value of UPSCC quantitative performance in sample cities is 0.64, and the values for 19 of the 35 sample cities are greater than or equal to the mean value, namely, Beijing, Nanjing, Hangzhou, Taiyuan, Guangzhou, Shanghai, Chengdu, Jinan, Wuhan, Lanzhou, Dalian, Changchun, Qingdao, Guiyang, Haikou, Shenyang, Xi’an, Urumqi and Xiamen. The bottom five cities in the UPSCC qualitative performance are Fuzhou, Shijiazhuang, Chongqing, Harbin and Xi’ning.

#### 5.2.3. Qualitative–Quantitative Bi-Dimensional (QQBD) Carrying Capacity of Urban Public Services

By classifying the performance value in Table 5 and Table 6 according to the analysis framework in Section 4.3, the UPSCC performance quadrant of the sample cities can be obtained as shown in Table 7.

The classification results of UPSCC performance in different dimensions and overall scores are shown in Table 7. According to Table 7, there are 9, 9, 8, 13, 10, 8 and 12 cities in quadrant I of the D1, D2, D3, D4, D5, D6 and D7 dimensions, respectively. The number of quadrant II cities is 8, 7, 10, 7, 6, 12, 3 in the D1, D2, D3, D4, D5, D6, and D7 dimensions, respectively. There are 10, 8, 8, 6, 10, 5, 11 quadrant III cities in the D1, D2, D3, D4, D5, D6, and D7 dimensions, respectively. There are 8, 11, 9, 9, 9, 9, 10, 9 cities in quadrant IV in the D1, D2, D3, D4, D5, D6, and D7 dimensions, respectively. In terms of overall performance, the number of cities in quadrants I, II, III, and IV are 11, 7, 9 and 8, respectively. Both the performance of UPSCC qualitative and UPSCC quantitative are good in quadrant I cities, such as Taiyuan, Dalian, Distributed, Shanghai, Jinan, Qingdao, Lanzhou, Lanzhou, Xiamen and Urumqi, and Hangzhou. Cities located in quadrant III have poor performance in both quantity and quality performance of public services, for example, Tianjin, Shijiazhuang, Hefei, Zhengzhou, Changsha, Shenzhen, Chongqing, Harbin. Meanwhile, cities in quadrants II and IV have poor performance in quality UPSCC and quantity UPSCC, respectively.

## 6. Discussion

### 6.1. Effectiveness of QQBD UPSCC Evaluation Method

Public service carrying capacity is an important perspective for measuring the relationship between public service supply and demand. Public service is a human-oriented social resource; therefore, not only quantity, but also quality should be considered when measuring UPSCC performance. In this study, a measurement model of public service carrying capacity from the perspective of quantity and quality is proposed to reflect the performance of public service. The following key issues affect the application of the proposed UPSCC evaluation method. First of all, the quantitative evaluation of public service carrying capacity needs to consider both UPSCC load and UPSCC carriers. Secondly, the qualitative evaluation of public service carrying capacity needs to carry out extensive social surveys and measure residents’ satisfaction with public service quality from the perspective of residents’ perception. Finally, qualitative and quantitative evaluation of public service carrying capacity requires a set of data. Through comparative analysis, cities with good performance and poor performance can be found, so as to improve in a targeted manner.

### 6.2. Interesting Findings from Case Demonstration

According to Table 5, Table 6 and Table 7, both the performance of UPSCC qualitative and UPSCC quantitative are good in some cities, which are located in quadrant I, such as Taiyuan, Dalian, Distributed, Shanghai, Jinan, Qingdao, Lanzhou, Lanzhou, Xiamen and Urumqi, and Hangzhou. From the perspective of UPSCC qualitative, the supply of public services is more satisfactory, and residents are more satisfied with public services. When referring to UPSCC quantitative, the urban public service carriers can bear the pressure caused by urban public services. Two quadrant I cities, Shanghai and Hangzhou, are taken as an example of the quantitative and qualitative performance of Shanghai’s public service carrying capacity. According to the study by Wang et al. (2020), Shanghai and Hangzhou are able to provide a lot of funds to improve the relevant public services, so that their public services are in a better position both in terms of quality and quantity [66]. Fang and Yu (2017) also opined that these better-developed cities have an important political and economic status, and they are given more resources for building more public service facilities [67].

Some cities located in quadrant III have poor performance in both quantity and quality performance of public services, for example, Tianjin, Shijiazhuang, Hefei, Zhengzhou, Changsha, Shenzhen, Chongqing, Harbin. In terms of the UPSCC quantity of these cities, it is difficult for carriers to bear the load pressure. In terms of UPSCC quality, the satisfaction with public service supply is low, and residents are not satisfied with the quality of public service. These cities have a strong attraction to population inflows for working, attending meetings, enjoying entertainment events, and traveling, and thus these cities present a surplus supply of public services [41].

Some cities have good public service quality, but poor public service quantity, and these are quadrant II cities, such as Ningbo, Hohhot, Fuzhou, Nanchang, Kunming, Xining and Yinchuan. In terms of quality, the supply of public services is more satisfactory, and residents are satisfied with public services. The public services provided by these cities have been unanimously recognized by residents, but the quantity of public services provided is insufficient. In terms of the public service quality of these cities, the carriers cannot bear the load pressure. This is because most of these cities are capitals of central and western provinces, and their economic development is relatively slow, so that the quantity of public service supply cannot meet the requirements of load. For example, the study by Liao et al. (2020) pointed out that the quality of public service in Kunming is very good, but the supply is insufficient [63].

The quantity of public services in some cities is good, but the quality is not good. These are the quadrant IV cities, for example, Guangzhou, Haikou, Xi’an, Beijing, Shenyang, Nanjing, Wuhan, Chengdu. In terms of public service quantity, the carriers can carry the load pressure. In terms of quality, the supply of public services is more satisfactory, and residents are more satisfied with the public services. For example, the quantity of public service carrying capacity of Beijing ranks first among the 35 cities. However, the quality score of its public service carrying capacity was only 413.3, lower than the average of all cities (429.723). In fact, the quantity of public services provided by Beijing is very abundant. According to the report of the Chinese Academy of Social Sciences (2020), the quantity of public service resources in Beijing ranks first in China. Previous studies have also indicated that the scale of public services in Beijing is sufficient. For example, Liu et al. (2015) considered that Beijing needs to improve the quality of public services [67].

## 7. Conclusions

This study proposed an innovation for evaluating urban public service carrying capacity from quantity and quality bi-dimensional perspectives. The conclusions from the results in this study can be drawn as follows. Firstly, public services are people-oriented social resources, which should be evaluated from both quantitative and qualitative perspectives. Secondly, the quantitative measurement of public service carrying capacity needs to consider the perspectives of load and carrier, while the qualitative measurement needs to consider the satisfaction of residents. Thirdly, the demonstration of the case cities showed the effectiveness of the qualitative–quantitative bi-dimensional UPSCC evaluation method.

The significance of this study can be highlighted as follows. The study contributes to the development of literature in the discipline of urban public carrying capacity. It offers an innovative method for evaluating the performance of UPSCC from a qualitative–quantitative bi-dimensional perspective. From a practical perspective, the application of the QQBD-based UPSCC method in the case demonstration can help local governments to identify weak areas of UPSCC. In other words, the deployment of the QQBD-based UPSCC method can help local governments to implement tailored measures to improve the performance of UPSCC.

There are limitations of this study that should be considered, in that only the data in 2019 of the 35 mega cities in China were used for empirical analysis to demonstrate the application of the QQBD-UPSCC method introduced in this study. It is recommended that the application of this method should be investigated on the basis of more cities. Further study also can extend the application of the QQBD evaluation method to other kinds of urban carrying capacity, such as infrastructure carrying capacity.

## Figures and Tables

**Figure 1 ijerph-18-12539-f001:**
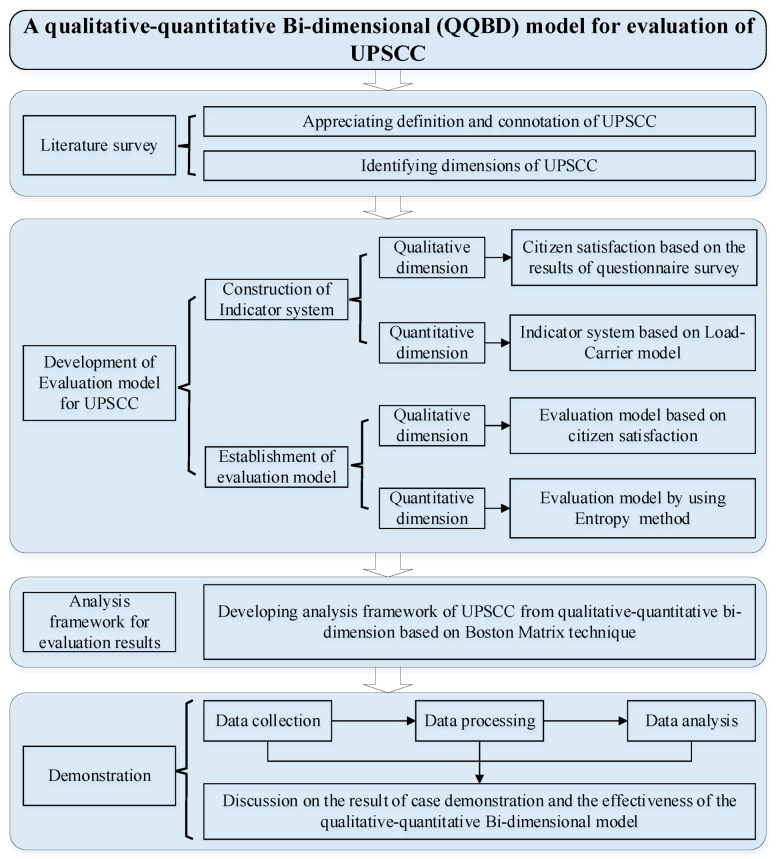
Research framework.

**Figure 2 ijerph-18-12539-f002:**
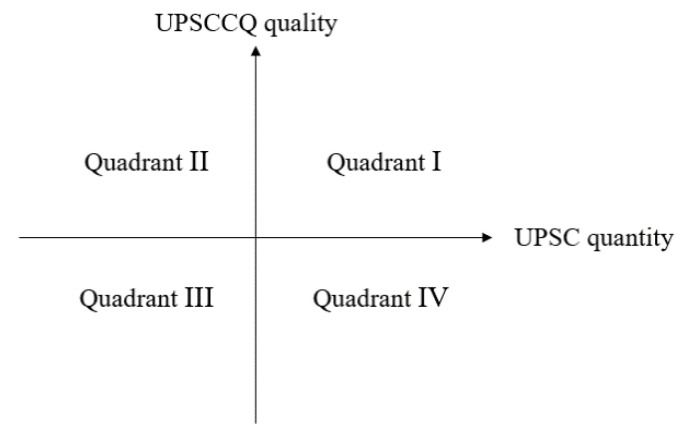
The analysis framework of UPSCC from qualitative–quantitative bi-dimension.

**Figure 3 ijerph-18-12539-f003:**
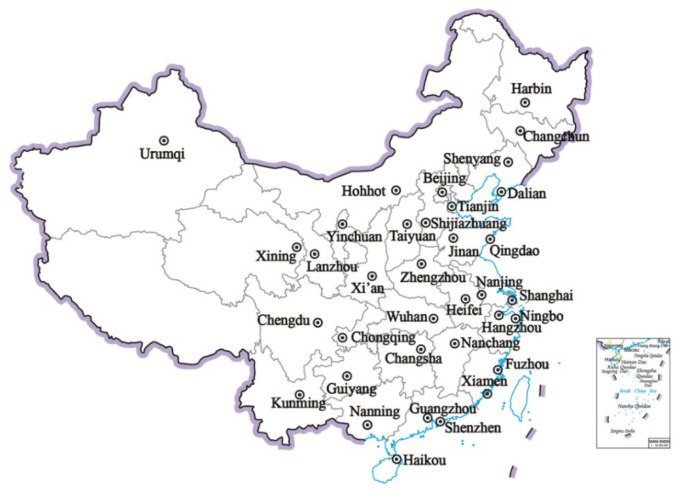
Spatial distribution of 35 sample cities.

**Table 1 ijerph-18-12539-t001:** Representative definitions of CC in the context of urban management.

References	Definition of CC in the Context of Urban Management
Godschalk and Parker, 1975	Carrying capacity is said to be the ability of natural and man-made systems to support the demands of various uses, and subsequently it refers to inherent limits in the systems beyond which instability, degradation, or irreversible damage occurs [34].
Schneider et al., 1978	Environmental carrying capacity is the ability of an ecosystem to sustain the human population without the natural or artificial environment system being severely degraded [35].
Oh et al., 2002	Urban carrying capacity is defined as a level of population growth, human activity, physical development and land us, which supports the urban habitat system to maintain sustainable development and does not cause its degradation and irreversible damage [36].
Oh et al., 2004	Urban carrying capacity concept in this research is defined as the maximum level of human activities which can be sustained by the urban environment without causing serious degradation and irreversible damage [37].
Liu and Borthwick, 2011	Carrying Capacity of the Environment is the combined threshold in time and space of natural resources, environmental assimilative capacity, ecosystem services, and social supporting capacity of the environment that could carry socio-economic activities without causing obvious changes or damage to structures and functions of the environment [28].
Ye et al., 2016	Resources and environmental carrying capacity is the social and economic pressure that can be placed on ecological surroundings on the condition that the ecosystem can maintain a stable structure and complete function [38].
Wang, 2016	The carrying capacity of urban public services refers to the maximum load and optimal scale of population and socio-economic activities that can be carried by various public services, within certain period of time and under the conditions of sustainable urban development [33].

**Table 2 ijerph-18-12539-t002:** Classification of dimensions of UPSCC in existing studies.

Reference	Dimension
OECD, 2011 [44]	General public service; defense; public order and safety; economic affairs; environmental protection; housing and community amenities; health; recreation; culture and religion; education; social protection
Afonso and Fernandes, 2008 [45]	Municipal expenditures; social services; basic education; cultural services; sanitation; territory organization; road infrastructure;
Andrews and Brewer, 2013 [46]	education; health; corrections; police protection; highways
Andrews and Entwistle, 2010 [47]	Education; social services; housing; highways; public protection; benefits and revenues
Stastna and Gregor, 2011 [48]	Public expenditures; administrative management; culture; education; urban environment; housing and industrial; public safety
Wei et al., 2015a [12]	basic municipal facilities; cultural and recreational facilities; sports facilities; educational facilities; housing conditions; healthcare facilities; public traffic
Sun et al., 2018 [49]	population; public health; education; technology innovation; public housing; transport infrastructure; industrial economy;
Domingues et al., 2015 [50]	Environmental protection and management; economic aspects; ethics and social responsibility
Weng et al., 2020 [51]	Economics; social; environment; transportation
Diao et al., 2019 [52]	Resource endowment; environmental protection; transportation; scientific and technological innovation; demographic factors; economic development
An and Ren, 2008 [53]	Social security; public security; public health; education; infrastructure; environmental protection; science and technology
Wang and Nan, 2011 [54]	Education; basic health care; public employment services and basic social security
Ma and Zeng, 2011 [55]	Education; public health; social security; infrastructure
Wei et al., 2015 [56]	Public security; education; culture; sports and media; social security; medical and health care; environmental protection; transportation
Li, 2011 [57]	Education; public health and basic medical care; basic social security and employment; public welfare infrastructure; public safety; environmental protection
Zhang, 2011 [58]	Education; basic medical and health care; social security; public employment; infrastructure; environmental protection; public security
Yang, 2015 [59]	Infrastructure; education; public employment; public health; social protection
Yin and Liu, 2017b [60]	Government expenditures on public services; education; medical and health care; scientific research and innovation; culture and sports; environmental protection; social security; and support for related industries
Wang et al., 2018 [23]	Compulsory education; medical and health care; social security; environmental governance; housing; population; economic development; household registration system; urban planning policies; public financial expenditures
Yang and Gu, 2019 [61]	Primary and secondary education; medical and health care; culture and sports; social welfare and security; basic life; roads and transportation; environmental protection; health and greening

**Table 3 ijerph-18-12539-t003:** Candidate indicators for evaluating the quantitative performance of UPSCC.

Dimension	Indicator
D1—Basic Education	Number of elementary school teachers per 10,000 people; number of middle school teachers per 10,000 people; number of elementary school per 10,000 people; number of junior high schools per 10,000 people; number of elementary school per administrative area; number of junior high schools per administrative area; school-age children enrollment rate; primary school promotion rate; middle school promotion rate; financial resources for education per pupil in primary schools; financial resources for education per student in junior high school; per capita state financial expenditure on education; ratio of education expenditure to regional financial expenditure; number of elementary school students per 10,000 people; number of junior high school students per 10,000 people; percentage of illiterate population aged 15 and over; number of students accommodated per junior high school; number of students accommodated per elementary school; number of middle school students per 10,000 people; number of elementary school students per 10,000 people; compulsory education expenditure per capita
D2—Public Health	Number of health facilities per 10,000 people; number of health facility personnel per 10,000 people; number of beds in medical institutions per 10,000 people; number of maternal and child health centers per million population; disease prevention and control per million population; number of health supervision offices per million population; hospital facility space per capita; average life expectancy per capita; medical institution bed use rate; per capita financial expenditure on health care; number of visits per unit of medical facility
D3—Social Security	proportion of basic pension insurance participants to total population; proportion of unemployment insurance participants to total population; proportion of basic medical insurance participants to total population; number of preferential placement units per 10,000 population; number of under-insurance per 100 people; unemployment rate; ratio of social security funding to fiscal spending; average wage; urban minimum living standard treatment; the amount of urban low income insurance payment per 10,000 people; minimum wage for employees; the minimum standard of living security for urban residents; older people as a proportion of the total population; social security spending per capita
D4—Housing security	Number of subsidized housing units per 10,000 people; Protected housing area per 10,000 people; Average price per square meter of housing; Average price of rent per square meter; House price to income ratio; Rent-to-income ratio; Housing security expenditure per capita
D5—Urban Environment	Number of wastewater treatment facilities per 10,000 people; number of abandoned treatment facilities per 10,000 people; sanitation vehicle population density; urban sewage treatment rate; harmless treatment rate of urban domestic waste; industrial “three wastes” treatment rate; greenery rate; green area per capita; air quality ratio; per capita financial expenditure on environmental protection; the proportion of environmental protection expenditure to fiscal expenditure; industrial wastewater emissions per 10,000 people; exhaust emissions per 10,000 people; industrial solid waste emissions per 10,000 people; domestic waste emissions per 10,000 people; respirable particulate matter concentration; sulfur dioxide concentration
D6—Public Transportation	Road area per capita; number of public transportation vehicles per 10,000 people; number of urban bus lines per 10,000 people; operating miles of urban rail transit per 10,000 people; average speed during peak hours; number of passengers carried per 100 buses; number of passengers carried per km of rail; number of cabs per 10,000 people; transportation expenditure per capita
D7—Public Culture and Sports	Number of cultural venues per 10,000 people; library population density; population density of public library collections; per capita financial expenditure on culture, sports and media

**Table 4 ijerph-18-12539-t004:** UPSCC carriers and load indicators used in this study.

Dimension	Carrier Indictor (Unit)	Load Indicator (Unit)
D1—Basic Education	C1 Number of primary schools (unit)	L1 Number of students in primary schools (person)
	C2 Number of teachers in primary schools (person)	L2 Number of students in primary schools (person)
	C3 Number of secondary schools (unit)	L3 Number of students in primary schools (person)
	C4 Number of teachers in secondary schools (person)	L4 Number of students in secondary schools (person)
	C5 Expenditure for education (10^4^ yuan)	L5 Resident population(person)
D2—Public Health	C6 Number of hospitals (unit)	L6 Visits of health institutions (10^4^ person-times)
	C7 Number of sickbeds (unit)	L7 Visits of health institutions (10^4^ person-times)
	C8 Number of physicians (person)	L8 Visits of health institutions (10^4^ person-times)
	C9 Financial investment in public health (10^4^ yuan)	L9 Resident population (person)
D3—Social Security	C10 Number of urban workers joining pension insurance (10^4^ person)	L10 Resident population (person)
	C11 Number of urban workers joining medical care insurance (104 person)	L11 Resident population (person)
	C12 Number of urban workers joining unemployment insurance (10^4^ person)	L12 Resident population (person)
	C13 Number of urban employee (10^4^ person)	L13 Number of residents receiving subsistence allowances in urban areas (10^4^ person)
D4—Housing Security	C14 Per capita consumption expenditure of urban households (yuan/person)	L14 Per capita housing expenditure of urban households (yuan/person)
	C15 Average annual wage of fully employed staff and workers (yuan/person)	L15 Average selling price of commercial building (yuan/m^2^)
D5—Urban Environment	C16 Treatment of wastewater (10^4^ m^3^)	L16Annual quantity of wastewater discharged (10^4^ m^3^)
	C17 Volume of harmless disposal of wastes (10^4^ ton)	L17 Municipal wastes collected and transported (10^4^ ton)
	C18 Area of parks and green land (hectare)	L18 Urban area (km^2^)
D6—Public Transportation	C19 Number of taxis (unit)	L19 Resident population (person)
	C20 Number of buses and trolley buses under operation (unit)	L20 Resident population (person)
	C21 Area of paved roads (10^4^ m^2^)	L21 Resident population (person)
D7—Public Culture and Sports	C22 Total collections of public libraries	L22 Resident population (person)
	C23 Number of mass cultural centers	L23 Resident population (person)
	C24 Number of sport halls	L24 Resident population (person)

**Table 5 ijerph-18-12539-t005:** The value of qualitative UPSCC performance value for the sample cities.

	D1	D2	D3	D4	D5	D6	D7	D		D1	D2	D3	D4	D5	D6	D7	D
C1	56.12	64.45	57.72	48.16	64.14	58.59	64.12	413.30	C19	63.18	65.97	61.07	51.47	61.28	66.86	66.05	435.88
C2	57.99	63.00	56.18	50.83	63.80	63.68	62.99	418.47	C20	54.88	61.19	54.86	46.93	62.26	61.77	58.61	400.50
C3	55.44	62.40	55.43	49.81	61.42	62.55	60.47	407.52	C21	55.59	62.77	54.28	45.70	63.85	61.04	61.08	404.31
C4	63.48	66.94	61.67	51.14	64.49	61.90	65.58	435.20	C22	58.61	61.67	55.79	49.07	64.21	62.92	59.36	411.63
C5	61.91	65.84	61.66	52.42	66.39	59.97	64.84	433.03	C23	56.07	63.25	55.82	45.85	68.75	58.04	61.14	408.92
C6	59.71	63.75	56.64	50.84	63.42	62.09	66.00	422.45	C24	57.46	63.24	56.60	45.58	69.86	61.43	62.71	416.88
C7	65.49	66.94	62.36	52.80	67.90	63.11	67.51	446.11	C25	61.12	64.49	57.55	48.64	68.91	62.22	65.61	428.54
C8	62.41	65.93	60.69	51.15	65.46	60.87	66.17	432.68	C26	59.49	58.30	55.35	51.30	65.16	57.28	71.50	418.38
C9	59.96	65.50	60.09	50.99	63.62	58.20	64.29	422.65	C27	57.64	63.05	56.73	51.91	68.73	63.88	62.72	424.66
C10	61.13	67.24	61.40	51.25	68.93	64.94	67.30	442.19	C28	56.22	62.94	56.46	47.37	66.37	63.67	62.64	415.67
C11	56.01	63.88	57.78	47.78	72.47	66.13	64.74	428.79	C29	68.78	71.11	66.42	50.00	72.90	58.12	71.38	458.71
C12	60.35	67.92	61.04	48.52	71.59	64.69	67.37	441.48	C30	64.92	67.12	61.74	48.51	69.67	57.89	64.83	434.68
C13	68.26	70.61	66.54	52.86	63.31	65.99	70.42	457.99	C31	44.93	59.17	51.01	43.99	59.35	59.10	57.96	375.51
C14	60.15	64.96	58.38	48.56	68.89	62.86	62.19	425.99	C32	69.31	72.29	67.28	53.13	70.93	63.31	71.06	467.31
C15	62.73	64.76	59.06	50.98	74.60	63.01	64.67	439.81	C33	67.46	69.62	65.66	58.61	71.92	65.72	68.86	467.85
C16	68.77	70.82	65.98	52.11	65.13	64.60	71.36	458.77	C34	60.64	65.95	60.27	53.74	68.20	64.64	65.94	439.38
C17	62.74	64.63	61.55	55.80	65.88	62.21	64.60	437.41	C35	61.91	65.15	61.71	52.96	65.41	60.84	61.94	429.92
C18	64.10	66.78	62.08	52.11	68.69	59.84	64.13	437.73	Average	60.71	65.25	59.57	50.37	66.80	62.11	64.92	429.72

**Table 6 ijerph-18-12539-t006:** The value of quantitative UPSCC performance value for the sample cities.

	D1	D2	D3	D4	D5	D6	D7	D		D1	D2	D3	D4	D5	D6	D7	D
C1	0.19	0.15	0.22	0.02	0.09	0.06	0.08	0.81	C19	0.13	0.11	0.15	0.04	0.06	0.10	0.07	0.67
C2	0.16	0.07	0.11	0.04	0.05	0.11	0.06	0.59	C20	0.08	0.15	0.16	0.04	0.07	0.06	0.06	0.61
C3	0.09	0.10	0.08	0.06	0.07	0.04	0.07	0.52	C21	0.13	0.14	0.16	0.04	0.06	0.11	0.05	0.70
C4	0.15	0.15	0.15	0.04	0.07	0.12	0.07	0.75	C22	0.09	0.13	0.15	0.04	0.06	0.07	0.06	0.61
C5	0.06	0.11	0.08	0.04	0.07	0.11	0.07	0.55	C23	0.11	0.12	0.21	0.01	0.09	0.11	0.05	0.72
C6	0.13	0.11	0.12	0.05	0.07	0.11	0.06	0.65	C24	0.09	0.07	0.23	0.02	0.09	0.10	0.04	0.64
C7	0.15	0.13	0.14	0.04	0.05	0.11	0.07	0.69	C25	0.07	0.13	0.09	0.06	0.07	0.09	0.07	0.58
C8	0.17	0.11	0.14	0.04	0.04	0.11	0.07	0.67	C26	0.10	0.14	0.17	0.04	0.04	0.11	0.07	0.66
C9	0.15	0.05	0.05	0.05	0.06	0.09	0.06	0.50	C27	0.08	0.11	0.13	0.05	0.06	0.03	0.05	0.51
C10	0.17	0.10	0.20	0.03	0.08	0.05	0.07	0.71	C28	0.11	0.14	0.19	0.04	0.07	0.08	0.08	0.71
C11	0.15	0.13	0.18	0.04	0.07	0.13	0.08	0.77	C29	0.12	0.13	0.16	0.04	0.07	0.08	0.07	0.67
C12	0.13	0.15	0.21	0.03	0.07	0.10	0.08	0.77	C30	0.11	0.15	0.10	0.04	0.05	0.06	0.07	0.59
C13	0.12	0.10	0.18	0.04	0.07	0.03	0.07	0.61	C31	0.11	0.13	0.14	0.04	0.05	0.11	0.07	0.65
C14	0.09	0.11	0.12	0.04	0.06	0.09	0.06	0.57	C32	0.16	0.13	0.13	0.04	0.05	0.13	0.06	0.70
C15	0.09	0.09	0.15	0.03	0.07	0.05	0.07	0.54	C33	0.10	0.15	0.00	0.04	0.07	0.09	0.05	0.50
C16	0.05	0.09	0.21	0.03	0.07	0.13	0.07	0.64	C34	0.04	0.13	0.12	0.05	0.07	0.12	0.07	0.60
C17	0.09	0.11	0.14	0.04	0.05	0.07	0.06	0.56	C35	0.08	0.12	0.16	0.04	0.03	0.14	0.07	0.65
C18	0.13	0.13	0.18	0.03	0.06	0.12	0.05	0.70	Average	0.11	0.12	0.15	0.04	0.06	0.09	0.07	0.64

**Table 7 ijerph-18-12539-t007:** The UPSCC performance quadrant for the sample cities.

Dimension	D1	D2	D3	D4	D5	D6	D7	D	Dimension	D1	D2	D3	D4	D5	D6	D7	D
Beijing	IV	IV	IV	III	IV	III	IV	IV	Qingdao	I	II	I	I	III	I	I	I
Tianjin	IV	III	III	II	III	I	III	III	Zhengzhou	III	IV	IV	III	IV	III	III	III
Shijiazhuang	III	III	III	IV	IV	II	IV	III	Wuhan	IV	IV	IV	III	III	IV	III	IV
Taiyuan	I	I	II	I	IV	IV	I	I	Changsha	III	IV	IV	IV	III	II	III	III
Hohhot	II	II	II	I	IV	IV	IV	II	Guangzhou	III	IV	IV	III	I	IV	III	IV
Shenyang	IV	III	III	I	IV	IV	II	IV	Shenzhen	III	III	IV	III	I	IV	III	III
Dalian	I	I	II	I	II	I	I	I	Nanning	II	IV	III	IV	I	II	I	III
Changchun	I	II	II	I	III	IV	I	I	Haikou	III	IV	IV	II	III	II	I	IV
Harbin	IV	II	II	I	III	III	III	III	Chongqing	III	III	III	I	II	II	III	III
Shanghai	I	II	I	II	I	II	I	I	Chengdu	IV	IV	IV	IV	IV	II	IV	IV
Nanjing	IV	IV	IV	IV	I	I	IV	IV	Guiyang	I	I	I	IV	I	III	I	I
Hangzhou	IV	I	I	III	I	I	I	I	Kunming	II	I	II	IV	II	III	IV	II
Ningbo	I	II	I	I	IV	II	I	II	Xi’an	III	IV	III	IV	III	IV	IV	IV
Hefei	III	III	III	IV	II	II	III	III	Lanzhou	I	I	II	I	II	I	II	I
Fuzhou	II	III	III	II	I	II	IV	II	Xining	II	I	II	I	I	II	II	II
Xiamen	II	II	I	II	IV	I	I	I	Yinchuan	III	I	II	I	I	I	I	II
Nanchang	II	III	II	II	III	II	III	II	Urumqi	II	IV	I	I	III	IV	IV	I
Jinan	I	I	I	II	II	IV	III	I									

## Data Availability

Not applicable.

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
