# Peer review of "Evaluation Method for Urban Public Service Carrying Capacity (UPSCC): A Qualitative–Quantitative Bi-Dimensional Perspective"

_ijerph, 2021, doi:10.3390/ijerph182312539_

Round 1

Reviewer 1 Report

The article is very interesting,  necessary and  scientifically mature. In fact, the only fundamental remark is that the key part of the article - Tables 6, 7 and 8 do not comment on the results of the calculations presented in them. What is in the conclusions and discussions is not enough with the commentary on the tables.
In addition, minor remarks:
line 18 - what does "perspective of load" mean?
line 19 - among stakeholders, administration and guess is still missing?
line 20 - instead of "case cities" maybe it's better to "laboratory cities" or "case study cities"?

thank you and good luck!

Author Response

No

Comment

Response

1

The article is very interesting, necessary and scientifically mature.

Thanks for the reviewer’s very supportive comments.

2

In fact, the only fundamental remark is that the key part of the article - Tables 6, 7 and 8 do not comment on the results of the calculations presented in them. What is in the conclusions and discussions is not enough with the commentary on the tables.

Thanks for the reviewer’s very helpful comments.

More comments have been added after Table 6, 7 and 8 to support the conclusions and discussions section.

The details can be seen in the revised manuscript.

3

In addition, minor remarks: line 18 - what does "perspective of load" mean?

line 19 - among stakeholders, administration and guess is still missing?

line 20 - instead of "case cities" maybe it's better to "laboratory cities" or "case study cities"?

thank you and good luck!

Thanks for the reviewer’s very constructive comments.

In line with the reviewer’s suggestion, all the mentioned remarks have been improved, shown as follows.

1. “perspective of load” in line 18 has been revised into “both UPSCC load and UPSCC carrier”

2. “among stakeholders, administration” has been added in line 19.

3. “case cities" has been revised into “case study cities” in line 20.The details can be seen in the revised manuscript.

Reviewer 2 Report

The manuscript addresses a very interesting topic and provides a methodological framework for regional development policy. The introduction of qualitative perspective in the model and the idea of dividing carrier and load seem relevant and interesting. I am interested in utilizing the method suggested by the authors and would consider it for my own research. It seems that the authors have put huge efforts for literature survey and framework design.

However, there are few weaknesses. I am still not sure how carrier and load indicators were treated in the calculation. If the authors could kindly describe further details of the treatment, It would be very helpful for readers.

In addition, It would be better if the authors provide general information about the cities. In the current version, there are only names of cities. General information such as population, GRDP, and main indutries etc are needed (if available).

Author Response

No

Comment

Response

1

The manuscript addresses a very interesting topic and provides a methodological framework for regional development policy. The introduction of qualitative perspective in the model and the idea of dividing carrier and load seem relevant and interesting. I am interested in utilizing the method suggested by the authors and would consider it for my own research. It seems that the authors have put huge efforts for literature survey and framework design.

Thanks for the reviewer’s very supportive comments.

2

However, there are few weaknesses. I am still not sure how carrier and load indicators were treated in the calculation. If the authors could kindly describe further details of the treatment, It would be very helpful for readers.

Thanks for the reviewer’s very kind comments.

In line with the reviewer’s suggestions, the following revisions have been made to show the calculation processing of carrier and load indicators.

1. Rename the coding number of UPSCC load in Table 5, to show the load for corresponding carrier more clearly.

2. A new section 3.4.3 has been added to show the detail of calculation process.

The details can be seen in the revised manuscript.

3

In addition, It would be better if the authors provide general information about the cities. In the current version, there are only names of cities. General information such as population, GRDP, and main indutries etc are needed (if available).

Thanks for the reviewer’s very meaningful comments.

In line with the reviewer’s suggestions, more general information about the case study cities (such as population, GRDP, and main industries) has been added in Section 5.1 and Appendix A.

The details can be seen in the revised manuscript.

Round 2

Reviewer 2 Report

The issues raised in the previous round review have well been addressed. I recommend this manuscript for publication in the journal.